

# Berry curvature and quantum oscillation from multi-orbital coadjoint-orbit bosonization

Mengxing Ye[1][⋆] and Yuxuan Wang[2][†]

**1** Department of Physics and Astronomy, University of Utah, Salt Lake City, UT, USA
**2** Department of Physics, University of Florida, Gainesville, FL, USA

⋆ mengxing.ye@utah.edu , † yuxuan.wang@ufl.edu

## Abstract

We develop an effective field theory for a multi-orbital fermionic system using the method of coadjoint orbits for higher-dimensional bosonization. The dynamical bosonic fields are single-particle distribution functions defined on the phase space. We show that when projecting to a single band, Berry curvature effects naturally emerge. In particular, we consider the de Haas-van Alphen effect of a 2d Fermi surface, and show that the oscillation of orbital magnetization in an external field is offset by the Berry phase accumulated by the cyclotron around the Fermi surface. Beyond previously known results, we show that this phase shift holds even for interacting systems, in which the single-particle Berry phase is replaced by the static anomalous Hall conductance. Furthermore, we obtain the correction to the amplitudes of de Haas-van Alphen oscillations due to Berry curvature effects.

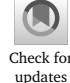

## 1   Introduction

The quantum oscillation of magnetization in the presence of an external field, or the de Haas-van Alphen effect (dHvA) plays an important role in modern condensed matter physics in characterizing the Fermi surface of gapless fermionic systems, which is commonly used to experimentally probe the size, shape, [1] and Berry phases [2–17] of the Fermi surface (FS).

For non-interacting systems, the dHvA effect can be obtained by summing over contributions from all Landau levels. An alternative, somewhat more intuitive way to understand the dHvA effect comes from a Bohr-Sommerfeld quantization of the semiclassical wavepacket moving around the FS, leading to oscillations as a periodic function of $\mathcal{A}_{FS}/B$ [18–20]. From this picture, an important outcome is that, for a multi-orbital system, the Berry phase $\gamma$ accumulated around the FS enters the dHvA oscillations as a phase shift [2, 14, 21], such that the free energy contains the components

$$F_k = A_k \cos\left[k\left(\frac{\mathcal{A}_{FS}}{B} + \gamma\right)\right], \quad k \in \mathbb{Z},$$

from which the magnetization $M = -\partial F/\partial B$ is obtained.

On the other hand, the effect of Berry curvature on the amplitudes $A_k$ has not been explicitly addressed. Furthermore, the Berry-phase induced phase shift has only been obtained using a single non-interacting wave packet. A natural question is how the phase shift $\gamma$ should be interpreted and renormalized in the presence of interaction effects. For interacting systems such as Fermi liquids and non-Fermi liquids, it becomes necessary to analyze this effect in a many-body field theory. However, traditional field-theoretic techniques can be clumsy in treating dHvA. First, due to the essential singularity in the $B$ dependence of magnetization, dHvA cannot be obtained in regular linear (or nonlinear) response theory; instead one needs to directly evaluate the free energy [22], which for a general lattice system is nontrivial to solve beyond the semiclassical regime [13, 23–27]. Second, for interacting systems, especially in 2d, the fermionic self-energy by itself contains oscillatory parts, whose closed-form expressions are not known [22, 28, 29]. It is thus highly desirable to develop an low-energy effective field theory in which Landau level physics near the Fermi level naturally emerge, and in which interaction effects can be further incorporated.

To this end, we develop a bosonization approach to study Berry phase effects in dHvA of a multi-orbital metal. As recently shown [30] using the method of coadjoint orbits [31] for higher-dimensional bosonization [31–40], an additional advantage of bosonization is that it reduces the problem of dHvA from one of *field theory* to one of *quantum mechanics*. In the bosonic theory, the base manifold is the *phase space* of single particles and the target manifold is the single-particle distribution function $f(\boldsymbol{x}, \boldsymbol{p})$ (where $f$ is a matrix for a multi-orbital system) under canonical transformations, i.e., the coadjoint orbit. While in this work we focus on free fermions, the interaction effects can be incorporated in the form of Landau parameters or coupling to bosonic collective modes [31], without going beyond Gaussian level, which is a major advantage of bosonization.

We focus on 2d systems with a fixed chemical potential $\mu$ (e.g., via gating) in a weak magnetic field ($B \ll k_F^2$, $k_F$ being the Fermi wavevector). We derive the multi-orbital bosonic theory from coherent-state path integral for the fermion bilinear field $\hat{f}_{\alpha\beta} = \hat{c}_\alpha^\dagger \hat{c}_\beta$. We couple the theory to an external magnetic field, and subsequently project this action via a gradient expansion onto a single band with a FS (see Fig. 1), and analyze the Berry curvature effects to leading order in $B$. We show that the Berry curvature modifies both the Poisson bracket and phase-space integration measure [25, 41].

By perturbatively expanding the action, we obtain a bosonzied theory of $N_\Phi$ chiral bosons $\phi_i$, where $N_\Phi$ is the Landau level degeneracy, in momentum space parametrized by the angle $\theta$. As we recently pointed out in [30], the dHvA effect comes from an additional total derivative term in $\phi_i(\theta)$ in the action. While not entering the equation of motion, this term becomes a topological $\theta$-term upon mode expansion, and has a nonperturbative effect on the quantization of energy. For multi-orbital systems with a single FS, we show that the Berry phase $\gamma$ enters the coefficient of the $\theta$-term, and thus leads to a phase shift of dHvA oscillations. Other than the phase shift, our field-theoretic approach allows for the calculation of the amplitudes $A_k$ of dHvA for all oscillatory components, and we obtain a Lifshitz-Kosevich-like formula [19] for $A_k$, which is modified by Berry curvature effects.

Remarkably, we show that the phase shift holds even for interacting systems. Via a nonperturbative proof that does not depend on the Hamiltonian, we show that the phase shift in dHvA is directly related to the static Hall conductance $\sigma^\mathrm{H}$. This relation can be directly tested in strongly correlated metals.

The rest of this paper is organized as follows. In Sec. 2 we present the multi-orbital coadjoint-orbit action. In Sec. 3 we couple the theory to a weak magnetic field and, after projection to a single band with a Fermi surface, obtain the bosonic action and equation of motion modified by multi-band effects. In Sec. 4 we perform a mode expansion along the Fermi surface, isolate the winding sector, and show how a topological $\theta$-term fixes the phase of dHvA. In Sec. 5 we quantize the theory at finite $T$ and use Poisson resummation to derive a Lifshitz–Kosevich–type formula with Berry-curvature corrections to both the phase and the amplitude. In Sec. 6 we relate the phase shift non-perturbatively to the static anomalous Hall conductance via the Kac–Moody algebra, and comment on its applicability in the presence of interactions. We conclude in Sec. 7 with a brief outlook. In Appendix A, we present the procedure for band projection to leading order in the gradient expansion for a generic Hamiltonian $\widehat{H}(\boldsymbol{x}, \boldsymbol{p})$. In Appendix B, we consider the case of a generic anisotropic FS, complementing the main text which assumes a circular one. In Appendix C, we detail the derivation of the modified Lifshitz-Kosevich formula, including the effect of deviation of the dispersion from a parabolic one. In Appendix D, we present a detailed discussion on the relation between the dHvA phase shift and the anomalous Hall conductance in the presence of interactions.

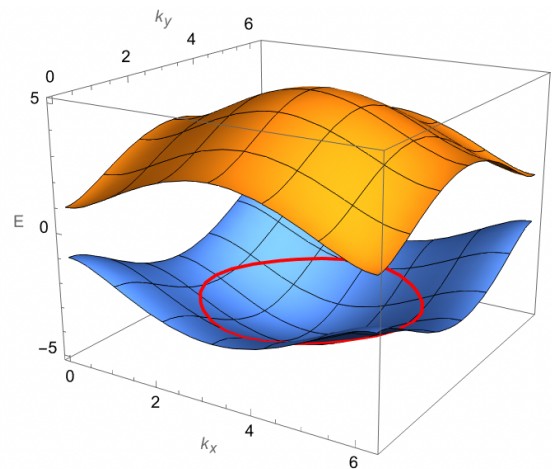

Figure 1: Illustration of the FS from a multi-orbital system, marked by the thick red line, enclosing a Fermi volume and a Berry phase.

## 2  Bosonized action for a multi-orbital system

For a non-interacting multi-orbital system, one can derive the bosonized action using coherent-state path integral [30,42–45]. As a general feature for bosonization, interacting effects can be straightforwardly incorporated [31]. Defining the ground state of the many-body Hamiltonian as $|\Omega\rangle$, we consider a coherent state given by $|\mathcal{U}\rangle = \hat{\mathcal{U}}|\Omega\rangle$, where $\hat{\mathcal{U}} = \exp\big(i\phi_{i\alpha,j\beta}\,c_{i\alpha}^\dagger c_{j\beta}\big)$ is parametrized by a bilocal field $\phi_{i\alpha,j\beta}$ that depends on both coordinates $i,j$ and orbital indices $\alpha,\beta$. The $\hat{\mathcal{U}}$'s, forming a Lie group, will be the target manifold of the field theory. Their action on the the ground state $|\Omega\rangle$ is known as the coadjoint orbit [31]. By standard derivations, the path integral is

$$Z = \int \mathcal{D}\hat{\mathcal{U}} \exp\left[\int dt\, \langle\Omega|\hat{\mathcal{U}}^{-1}\big(-\partial_t - i\hat{\mathcal{H}}\big)\hat{\mathcal{U}}|\Omega\rangle\right], \tag{1}$$

where $\mathcal{D}\hat{\mathcal{U}}$ is the product of the Haar measure of the Lie group taken at all times, and $\hat{\mathcal{H}}$ is the multi-orbital noninteracting Hamiltonian. The operators in the action can be expanded in terms of nested commutators of fermion bilinear operators, which can be represented by first-quantized operators. The action then becomes $S = \int dt\, \mathrm{Tr}\big[\widehat{f_0}\widehat{U}^{-1}i\partial_t\widehat{U} - \widehat{f}\,\widehat{H}\big]$. where $\widehat{f_0}$ and $\widehat{f}$ are one-particle density matrices for the ground state and the coherent state, defined respectively as $f_{0,j\beta,i\alpha} \equiv \langle\Omega|c_{i\alpha}^\dagger c_{j\beta}|\Omega\rangle$ and $f_{j\beta,i\alpha} \equiv \langle\mathcal{U}|c_{i\alpha}^\dagger c_{j\beta}|\mathcal{U}\rangle$. Further, here $\widehat{U}$ and $\widehat{H}$ are first-quantized counterparts of $\hat{\mathcal{U}}$ and $\hat{\mathcal{H}}$, i.e., $\widehat{U} \equiv e^{i\widehat{\phi}}$, with $\langle i\alpha|\widehat{\phi}|j\beta\rangle = \phi_{i\alpha,j\beta}$ and the elements of $\widehat{H}$ are given by $\hat{\mathcal{H}} = c_{i\alpha}^\dagger H_{i\alpha,j\beta}c_{j,\beta}$. The trace over single-particle operators can be alternatively expressed as phase-space integrals over Moyal products [30,46] of Wigner functions, which leads to

$$S = \int dt\, \frac{d\boldsymbol{x}\,d\boldsymbol{p}}{(2\pi)^2} \mathrm{Tr}\big[\widehat{f_0}(\boldsymbol{x},\boldsymbol{p}) \star \widehat{U}^{-1}(\boldsymbol{x},\boldsymbol{p},t) \star i\partial_t\widehat{U}(\boldsymbol{x},\boldsymbol{p},t) - \widehat{f}(\boldsymbol{x},\boldsymbol{p}) \star \widehat{H}(\boldsymbol{x},\boldsymbol{p})\big], \tag{2}$$

where $\star \equiv \exp\big[i(\overleftarrow{\partial_{\boldsymbol{x}}}\overrightarrow{\partial_{\boldsymbol{p}}} - \overleftarrow{\partial_{\boldsymbol{p}}}\overrightarrow{\partial_{\boldsymbol{x}}})/2\big]$ is the Moyal product, and $\widehat{U}(\boldsymbol{x},\boldsymbol{p})$, $\widehat{f}(\boldsymbol{x},\boldsymbol{p}) = \widehat{U}(\boldsymbol{x},\boldsymbol{p}) \star \widehat{f_0}(\boldsymbol{x},\boldsymbol{p}) \star \widehat{U}^{-1}(\boldsymbol{x},\boldsymbol{p})$, and $\widehat{H}(\boldsymbol{x},\boldsymbol{p})$ are the Wigner functions of the corresponding operators, which are defined via, e.g., $\widehat{U}(\boldsymbol{x},\boldsymbol{p}) = \int d\boldsymbol{y}\, e^{i\boldsymbol{p}\cdot\boldsymbol{y}} \langle\boldsymbol{x}-\boldsymbol{y}/2|\widehat{U}|\boldsymbol{x}+\boldsymbol{y}/2\rangle$. All these Wigner functions are matrices in orbital space, over which the trace is taken in Eq. (2).

# 3 Band projection in a weak magnetic field

Without loss of generality, we consider a 2d multi-orbital metal with a single closed FS, sketched in Fig. 1, in the presence of a magnetic field. For weak fields, we have $Ba_0^2 \ll 1$ (where $a_0$ is the lattice constant), and following the Peierls substitution, the Hamiltonian can be written as $\widehat{H}(\boldsymbol{x}, \boldsymbol{p}) = \widehat{H}(\boldsymbol{\pi} = \boldsymbol{p} + \boldsymbol{A}(\boldsymbol{x}))$, where $\boldsymbol{\pi}$ is the kinetic momentum. As a low-energy effective field theory (EFT), we can project the theory onto a single band $|v_{\boldsymbol{\pi}}\rangle$ [47–49]. Since $\widehat{H}(\boldsymbol{\pi})$ is treated as a Wigner function instead of a quantum operator, the proper multiplication operator is $\star$ (which satisfies associativity), i.e., the energy eigenstate is given by $\widehat{H} \star |v\rangle = |v\rangle \star \epsilon$, with $\langle v| \star |v\rangle = 1$. The effective action is

$$S_{\text{eff}} = \int_{t, \boldsymbol{x}, \boldsymbol{p}} \langle v| \star \left( \widehat{f_0} \star \widehat{U}^{-1} \star i \partial_t \widehat{U} - \widehat{f} \star \widehat{H} \right) \star |v\rangle \,. \tag{3}$$

Here $\langle v_{\boldsymbol{\pi}}| \star \widehat{f_0}(\boldsymbol{\pi}) = f_0(\boldsymbol{\pi}) \star \langle v_{\boldsymbol{\pi}}|$, where $f_0(\boldsymbol{\pi}) = \theta(\epsilon_F - \epsilon(\boldsymbol{\pi}))$ up to linear order in gradient expansion of the Moyal product [50] and the low-energy fluctuations are restricted to intra-band ones: $\langle u| \star \widehat{U} \star |v\rangle = U \delta_{uv}$. (See Appendix A for more details.)

For low-energy physics, we will expand our theory around the FS. Thus, for a closed FS, one can ignore the BZ boundary and take the continuum limit. In this situation, as pointed out in Ref. [30], it is convenient to work in the $(\boldsymbol{R}, \boldsymbol{\pi})$ coordinates of the phase space, where $\boldsymbol{R} = (x - \pi_y/B, y + \pi_x/B)$ is the guiding center, as the theory is $\boldsymbol{R}$-independent due to magnetic translation symmetry. In this situation, the Moyal product simplifies to $\star = \exp\left[ -i(B/2)\overleftarrow{\partial_{\boldsymbol{\pi}}} \times \overrightarrow{\partial_{\boldsymbol{\pi}}} \right]$ [30]. Noting that $|v\rangle$ is *not* the eigenstate of $H$ via matrix multiplication, let us assume instead $H|v_0\rangle = \epsilon_0 |v_0\rangle$, where $\langle v_0|v_0\rangle = 1$. We can expand $|v\rangle$ as $|v_0\rangle + |\delta v\rangle + \mathcal{O}(B^2)$. We have $1 = \langle v| \star |v\rangle \approx \langle v_0| \star |v_0\rangle + 2\,\text{Re}\,\langle v_0|\delta v\rangle + \mathcal{O}(B^2) = 1 - B\Omega/2 + 2\,\text{Re}\,\langle v_0|\delta v\rangle + \mathcal{O}(B^2)$, where $\Omega(\boldsymbol{\pi}) = i \langle v_0| \overleftarrow{\partial_{\boldsymbol{\pi}}} \times \overrightarrow{\partial_{\boldsymbol{\pi}}} |v_0\rangle$ is the Berry curvature. We see that $\text{Re}\,\langle v_0|\delta v\rangle = B\Omega/4 + \mathcal{O}(B^2)$. In the weak field limit, $\mathcal{O}(B^2)$ terms are suppressed by $B/k_F^2$, and can be safely neglected (which we do hereafter). Knowing the expansion of $|v\rangle$, we can then expand the projection of $\widehat{H}$ [23, 51]

$$\begin{aligned}
\langle v| \star \widehat{H}(\boldsymbol{\pi}) \star |v\rangle &= 2\,\text{Re}\,\langle v_0|H|\delta v\rangle + \langle v_0| \star H \star |v_0\rangle \\
&= \epsilon_0(\pi_x - B\mathcal{A}_y, \pi_y + B\mathcal{A}_x) - \mathcal{M}(\pi_x, \pi_y)B \,.
\end{aligned} \tag{4}$$

Here $\mathcal{M}(\boldsymbol{\pi}) = -\frac{i}{2} \langle \nabla_{\boldsymbol{\pi}} v_0| [H(\boldsymbol{\pi}) - \epsilon(\boldsymbol{\pi})] \times |\nabla_{\boldsymbol{\pi}} v_0\rangle$ is the spontaneous orbital magnetic moment, first obtained in a gauge invariant way using semiclassical wave packets [6], and $\mathcal{A}(\boldsymbol{\pi}) = i \langle v_0| \partial_{\boldsymbol{\pi}} |v_0\rangle$ is the Berry connection. We can thus define the modified kinetic momentum as

$$k_{x,y} \equiv \pi_{x,y} \mp B\mathcal{A}_{y,x} \,, \tag{5}$$

and the effects of Berry curvature enters through a Jacobian $J = 1 + B\Omega$ present both in the integration measure and in the Moyal product. Note that while it is tempting to treat $\boldsymbol{A}(\boldsymbol{x})$ and $\mathcal{A}(\boldsymbol{p})$ as components of phase-space gauge fields, a direct minimal coupling procedure $(\boldsymbol{x}, \boldsymbol{p}) \to (\boldsymbol{x} - \mathcal{A}, \boldsymbol{p} + \boldsymbol{A})$ is not correct (see Appendix A.3 for the complete analysis).

After the band projection, the action becomes

$$S = \int_{t, \boldsymbol{R}, \boldsymbol{k}} (1 + B\Omega) \left[ f_0(\boldsymbol{k}) U^{-1} \star i \partial_t U - f \star \epsilon(\boldsymbol{k}) \right] \,, \tag{6}$$

where $\star = \exp\left[ -iB\overleftarrow{\partial_{\boldsymbol{k}}} \times \overrightarrow{\partial_{\boldsymbol{k}}} / 2(1 + B\Omega) \right]$, and $f = U \star f_0(\boldsymbol{k}) \star U^{-1}$ in the new coordinates, and the effective dispersion is given by

$$\epsilon(\boldsymbol{k}) = \epsilon_0(\boldsymbol{k}) - \mathcal{M}(\boldsymbol{k})B \,. \tag{7}$$

The action (6) agrees with a recent effective field theory phenomenologically obtained from single-particle symplectic mechanics [41]. However, the spontaneous moment contributes a "Zeeman energy" $-\mathcal{M}(\boldsymbol{k})B$ to the dispersion, which cannot be obtained using the methods there, and has to be assumed to be implicit in $\epsilon(\boldsymbol{k})$. Notably, the saddle point of this action yields $\partial_t f(\boldsymbol{R}, \boldsymbol{k}, t) = \{f(\boldsymbol{R}, \boldsymbol{k}, t), \epsilon(\boldsymbol{k})\}_{\mathrm{M}}$, where $\{F, G\}_{\mathrm{M}} \equiv -i(F \star G - G \star F)$ is the Moyal bracket [30, 44, 46] and can be approximated by the Poisson bracket

$$\{F, G\} = -\frac{B \partial_{\boldsymbol{k}} F \times \partial_{\boldsymbol{k}} G}{(1 + B\Omega(\boldsymbol{k}))},$$

($\partial_{\boldsymbol{k}}$ taken with $\boldsymbol{R}$ fixed); indeed, the corrections due to Moyal product comes at $\mathcal{O}(B^3)$, an even higher order than the Berry curvature effects. Further defining $\boldsymbol{X} = \boldsymbol{x} + \mathcal{A}(\boldsymbol{\pi})$, using the chain rule we have in $(\boldsymbol{X}, \boldsymbol{k})$ coordinates (see also Appendix A.3)

$$\partial_t f(\boldsymbol{X}, \boldsymbol{k}, t) = \frac{-\boldsymbol{v}(\boldsymbol{k}) \cdot \partial_{\boldsymbol{X}} f - B\boldsymbol{v}(\boldsymbol{k}) \times \partial_{\boldsymbol{k}} f}{1 + B\Omega(\boldsymbol{k})}, \tag{8}$$

where $\boldsymbol{v} = \partial_{\boldsymbol{k}}\epsilon$ and $\partial_{\boldsymbol{k}} f$ is taken with $\boldsymbol{X}$ fixed. This is precisely the collisionless Boltzmann equation in the presence of Berry curvature and magnetic field. The $1 + B\Omega$ factor was first obtained from wavepackets [25], and has been shown to follow from band projection [47, 48, 52, 53], although as can be seen in Appendix A.3, our derivation using $(\boldsymbol{R}, \boldsymbol{k})$ coordinates is significantly more straightforward.

The phase-space field can be expressed as $U(\boldsymbol{R}, \boldsymbol{k}) = \exp(i\phi(\boldsymbol{R}, \boldsymbol{k}))$, where $\phi \simeq \phi + 2\pi$ is a real compact field. We expand the effective action up to quadratic order in $\phi$, and drop higher-order terms, which are irrelevant perturbations. When $\phi$ is single-valued, the quadratic action in the weak field limit reads

$$S^{(2)} = \int \frac{d\boldsymbol{R}d\boldsymbol{k}}{8\pi^2}(1 + B\Omega)\{\phi, f_0(\boldsymbol{k})\}\left(\dot{\phi} - \{\epsilon(\boldsymbol{k}), \phi\}\right). \tag{9}$$

Further, as shown in Ref. [30], due to magnetic translation symmetry, the integral over non-commutative (phase-space) coordinates $\boldsymbol{R}$ can be converted to a sum over $N_\Phi$ magnetic Bloch states in the magnetic Brillouin zone (mBZ), where $N_\Phi = BL^2/2\pi$ is the Landau level degeneracy: we have $\int d\boldsymbol{R}/(2\pi B^{-1})... = \mathrm{Tr}_{\widehat{R}}\,... = \sum_{K_i \in \mathrm{mBZ}} \langle K_i | ... | K_i \rangle$. In addition, it suffices [30] to only include in the path integral over $\phi(\boldsymbol{R}, \boldsymbol{k})$ that are invariant under magnetic translation, i.e., $\langle K_i | \phi(\widehat{\boldsymbol{R}}, \boldsymbol{k}) | K_j \rangle = \phi_i(\boldsymbol{k})\delta_{ij}$.

Let us assume the FS is isotropic, and show the analysis for anisotropic FS's in Appendix B. In terms of $\phi_i(\boldsymbol{k})$'s, the effective action is given by $S_{\mathrm{eff}} = \sum_{i=1}^{N_\Phi}(S_{\mathrm{cb}} + S_{\mathrm{w}})$, with

$$\begin{aligned}
S_{\mathrm{cb}} &= \int \frac{dt\,d\theta}{4\pi} \partial_\theta \phi_i \left(\dot{\phi}_i - \frac{\omega_c}{1 + B\Omega_\theta} \partial_\theta \phi_i\right), \\
S_{\mathrm{w}} &= \int \frac{dt\,d^2k}{2\pi} f_0(\boldsymbol{k}) \left[-\frac{1 + B\Omega}{B} \dot{\phi}_i + \frac{1}{2}(\partial_{\boldsymbol{k}} \times \partial_{\boldsymbol{k}} \phi_i) \dot{\phi}_i\right],
\end{aligned} \tag{10}$$

where the cyclotron frequency $\omega_c = Bv(k_F)/k_F$ and $\Omega_\theta = \partial_{\boldsymbol{k}} \times \mathcal{A}|_{k=k_F}$ are taken on the FS. The $S_{\mathrm{cb}}[\phi_i(\theta, t)]$ term directly follows from Eq. (9), where $\theta$ parametrizes the tangential direction along the Fermi surface. It describes a chiral boson propagating in 1d momentum space [54]. The $S_{\mathrm{w}}[\phi_i(\boldsymbol{k}, t)]$ term, commonly neglected in literature, is nonzero when $\phi_i$ contains winding configurations [30], and is topological (depending only on the winding numbers). As is well-known [55], topological terms do not enter the Hamiltonian, but they do affect the quantization of the theory.

# 4 Mode expansion and quantization

We perform a mode expansion for the compact field $\phi_i$ as

$$\phi_i(\theta) = q_i + p_i\theta + \sum_{n\neq 0}\frac{a_{i,n}}{\sqrt{|n|}}e^{in\theta}, \qquad \theta \in [-\pi,\pi),$$

where $q_i \simeq q_i + 2\pi$, and $p_i$ is a winding number. For $p_i \neq 0$, the $U_i = e^{i\phi_i(k)}$ field has a singular (vortex) configuration in $k$, which requires an extension of the coherent states $|\mathcal{U}\rangle$ in the path integral [30]. The physical meaning of $p_i$ is the extra occupation number at a given $K_i$. Indeed, from

$$\delta f_i = \int_k (1 + B\Omega)\left(e^{i\phi_i} \star f_0 \star e^{-i\phi_i} - f_0\right) = -\int_\theta \frac{\partial_\theta \phi_i}{2\pi}, \tag{11}$$

we see that adding $p_i \neq 0$ configurations for each $K_i$ ensures that the system is in a grand canonical ensemble (as it should for a fixed $\mu$).

To quantize $S_w$ we need to extend the mode expansion into the Fermi sea, but since the term is topological, the specific form of extension does not matter [30]. The action splits into a Fock sector and a zero-mode sector, $S_{cb}[\phi] + S_w[\phi] = S_{Fock}[a_n] + S_{zero}[p,q]$. As shown in Ref. [30], $S_{Fock}$ is responsible for specific heat and Landau diamagnetism for $T \gg \omega_c$, which we will not discuss here.

The zero-mode action can be written as

$$S_{zero}[p,q] = \int dt\left[\left(-\frac{\mathcal{A}_{FS}}{2\pi B} - \frac{\gamma}{2\pi}\right)\dot{q} + p\dot{q} - \frac{\bar{\omega}_c}{2}p^2\right], \tag{12}$$

where $\gamma = \int_k f_0(k)\Omega(k)$ is the Berry phase around the FS, and the effective cyclotron frequency for a generic FS is [25] (see Appendix B)

$$\bar{\omega}_c(B,\Omega) = \frac{B}{2\pi g(\epsilon_F,B)}\left\langle\frac{1}{1 + B\Omega_\theta}\right\rangle_{FS}, \tag{13}$$

where $g(\epsilon_F,B)$ is the density of states obtained from (7), and $\langle...\rangle_{FS}$ denotes an average over the FS for anisotropic cases.

Interestingly, the action (12) exactly maps to that of a quantum mechanical problem of a charged particle moving on a ring enclosing a flux, where the first term is a topological $\theta$-term [55]. At a finite temperature $1/\beta$, the partition function is

$$Z_{zero} = \sum_{p'\in\mathbb{Z}}\exp\left[-\frac{\beta\bar{\omega}_c}{2}\left(p' + \frac{\mathcal{A}_{FS}}{2\pi B} + \frac{\gamma}{2\pi}\right)^2\right]. \tag{14}$$

We clearly see that this result is periodic in $\mathcal{A}_{FS}/B + \gamma \mod 2\pi$ at low-temperatures, and becomes non-oscillatory at $T \gg \bar{\omega}_c$ when the summation can be replaced by integral. This is precisely the origin of dHvA.

# 5 Modified Lifshitz-Kosevich formula

The oscillation of orbital magnetization in an external field can be obtained by evaluating the free energy of the zero-mode partition function, $F_{osc} = -T\log Z_{zero}$. Using the Poisson

resummation formula,

$$
\begin{aligned}
F_{\text{osc}} &= -N_{\Phi} T \log\left[ 1 + 2 \sum_{n=1}^{\infty} \cos(n\Delta) q^{n^2} \right] \\
&= -2 N_{\Phi} T \,\text{Re} \sum_{m=1}^{\infty} \log\left( 1 + q^{2m-1} e^{i\Delta} \right) + \text{non-osc.},
\end{aligned}
$$

where $q = \exp\left(-2\pi^2 T/\bar{\omega}_c\right)$ and $\Delta = \mathcal{A}_{\text{FS}}/B + \gamma$. In going to the second line, we used a property of the Jacobi theta function $\theta_3(\Delta/2, q)$ [56]; see Appendix C for details. Expanding the log in the second line and performing the summation over $m$, we obtain

$$
F_{\text{osc}} = N_{\Phi} T \sum_{k=1}^{\infty} \frac{(-)^k \cos\left[k\left(\mathcal{A}_{\text{FS}}/B + \gamma\right)\right]}{k \sinh(2\pi^2 k T/\bar{\omega}_c)}. \tag{15}
$$

A key modification to previous results [12, 13] is that $\bar{\omega}_c$ in the oscillation amplitude is non-linear in $B$, due to both Berry phase and the magnetic moment. For $T \geq \bar{\omega}_c$, expanding in the exponent in $B$ using Eq. (13), we get for the amplitude:

$$
A_k \propto \exp\left(-\frac{k\lambda_1 T}{B}\right) \exp(k\lambda_2 T), \tag{16}
$$

where

$$
\lambda_1 = 4\pi^3 g(\epsilon_F, 0), \qquad \lambda_2 = 4\pi^4 g^2(\epsilon_F, 0) \frac{\partial^2 \bar{\omega}_c}{\partial B^2}\bigg|_{B=0}. \tag{17}
$$

The additional $\lambda_2$ term can be directly tested in 2d materials with strong Berry curvature on the Fermi surface. While the same result could be obtained via semiclassical wavepackets [13, 23–25], it has not been given explicitly. Moreover, our systematic procedure of expanding in $B$ and in $\phi$ allows for obtaining higher-order corrections. For example, we show in Appendix C.1 that including $\phi^3$ terms in the action lead to a small temperature-dependent phase shift in dHvA for non-parabolic bands, which was recently reported in 3d metals [57].

The periodicity and phase shift in (15) is rooted in the topological $\theta$-term in (12), and survives even for interacting systems such as Fermi liquids and non-Fermi liquids [31]. While the Luttinger theorem [58–60] ensures the physical meaning of $\mathcal{A}_{\text{FS}}$, a natural question is that of $\gamma$ beyond single-particle physics.

## 6 dHvA and the anomalous Hall effects

The canonical quantization of $S_{\text{cb}} + S_{\text{w}}$, from the $\sim \dot{\phi}$ terms therein, leads to the Kac-Moody algebra [30, 60, 61]

$$
[\hat{n}_i(\theta), \hat{n}_i(\theta')] = \frac{-i}{2\pi} \delta'(\theta - \theta'), \tag{18}
$$

where $n_i(\theta) \equiv -\partial_\theta \phi_i(\theta)/2\pi$ is the density of electrons with magnetic momentum $K_i$ (see Eq. (11)).

For a system without Berry phases, this algebra was used [60] to obtain a nonperturbative derivation of the Luttinger's theorem. In our context, let's set $N_{\Phi} = 1$ and consider the braiding algebra between "translation" operators generated by kinetic momentum $\pi$:

$$
\hat{\mathbb{T}}_{x,y} = \exp\left( i \sum_{\pi} \hat{N}_\pi \pi_{x,y} \right),
$$

where $\hat{N}_\pi$ is the particle number operator and we set lattice constant $a = 1$. From basic quantum mechanics (see Appendix D), $\hat{\mathbb{T}}_x \hat{\mathbb{T}}_y \hat{\mathbb{T}}_x^{-1} \hat{\mathbb{T}}_y^{-1} = e^{2\pi i \nu}$ [60], where $\nu$ is the filling fraction of the band. In the low-energy limit, the lattice translation operators become $\hat{\mathbb{T}}_{x,y} \sim \exp\{i \int_\theta \hat{n}(\theta)[k_{F;x,y}(\theta) \pm B\mathcal{A}_{y,x}(\theta)]\}$, where we have used (5). Together with (18), up to $\mathcal{O}(B^2)$ corrections, the same braiding algebra evaluates to (see Appendix D)

$$\hat{\mathbb{T}}_x \hat{\mathbb{T}}_y \hat{\mathbb{T}}_x^{-1} \hat{\mathbb{T}}_y^{-1} = \exp\left[i\frac{\mathcal{A}_{\mathrm{FS}}}{2\pi} + i\frac{B\gamma}{2\pi}\right], \tag{19}$$

where $\gamma \equiv \oint \mathcal{A} \cdot d\boldsymbol{k}$ is the same as that in (12) via Stokes theorem. Relating the right band sides, we get $4\pi^2 \nu = \mathcal{A}_{\mathrm{FS}} + B\gamma = \mathcal{A}_{\mathrm{FS}}^0 + (\partial_B \mathcal{A}_{\mathrm{FS}} + \gamma)B$, where $\mathcal{A}_{\mathrm{FS}}^0$ is the zero-field Fermi surface area corresponding to the Luttinger theorem, and the $\partial_B \mathcal{A}_{\mathrm{FS}}$ term is due to the spontaneous magnetization in Eq. (7). The linear charge response to a $B$ field can be related to the *static* (anomalous) Hall conductance via the Streda formula [41,62]:

$$\sigma^{\mathrm{H}}(q \to 0, \omega = 0) = \left.\frac{d\nu}{dB}\right|_{B=0} = \frac{\gamma}{4\pi^2} + \frac{\partial_B \mathcal{A}_{\mathrm{FS}}}{4\pi^2}. \tag{20}$$

This result has been long known for free fermions [25,63], and recently proven perturbatively for interacting systems via Green's functions [64]. Our non-perturbative derivation, only taking input from low energies,[1] applies even for interacting systems such as Fermi liquids and non-Fermi liquids [64,65]. We prove the applicability to interacting systems in Appendix D.1.

From (15) and (20), we get for a generic interacting system

$$F_{\mathrm{osc}} = \sum_{k=1}^{\infty} A_k \cos\left[k\left(\frac{\mathcal{A}_{\mathrm{FS}}^0}{B} + 4\pi^2 \sigma^{\mathrm{H}}\right)\right]. \tag{21}$$

Thus, the phase shift in dHvA should precisely match the static anomalous Hall conductance, which can be extracted from spatially-resolved transport [66].

# 7 Discussion

The key result of this work, Eq. (15), takes a similar form of the Lifshitz-Kosevich formula, with two important differences. First, the renormalized cyclotron frequency $\bar{\omega}_c$ in (13) is nonlinear in $B$, leading to a modified temperature dependence. Second, the phase shift $\gamma$ applies beyond the single-particle Berry phase for free fermions. We note that unlike the phase shift, the amplitude $A_k$ in general receives additional corrections from interaction effects, which we will address in an upcoming work [67]. Finally, it would be interesting to extend our results to 3d systems.

# Acknowledgments

We would like to thank Aris Alexandradinata, Jing-Yuan Chen, Andrey V. Chubukov, Luca V. Delacrétaz, Dominic Else, Eduardo Fradkin, and Dmitrii L. Maslov for useful discussions. We especially thank Leon Balents for helpful discussions on band projection via a gradient expansion.

**Funding information** YW is supported by NSF under award number DMR-2045781. MY is supported by a start-up grant from the University of Utah. We acknowledge support by grant NSF PHY-1748958 to the Kavli Institute for Theoretical Physics (YW and MY), where this work was partly performed.

---

[1]*A priori* one cannot use $\nu = \int f_0(p) dp / 4\pi^2$ except for free fermions.

In the Appendix we provide details on (A) the band projection procedure for a generic Hamiltonian $\widehat{H}(\boldsymbol{x}, \boldsymbol{p})$ (B) the analysis for anistropic FS's, (C) the derivation of the modified Lifshitz-Kosevich formula from bosonization, and (D) the relation between the dHvA phase shift and anomalous Hall conductance.

## A  Band projection procedure for general phase space coordinates

In this section, we generalize the band projection procedure discussed in the main text (Eq. (4)) to a general set of phase space coordinates. We show that while the action derived in Eq. (6) of the main text remains the same using different choices of phase space coordinates, proper coordinates can be chosen to simplify the calculation. Furthermore, the procedure can be readily applied to electromagnetic responses, and to the leading order in the gradient expansion, the electromagnetic responses *only* depend on the Berry curvature and the orbital magnetic moment.

The Moyal product in a set of phase space coordinates $\{\xi^i\}$ with $i = 1, 2, \ldots, 2d$ reads

$$F \star G = F \exp\left\{ \frac{i\hbar}{2} \omega_\xi^{ij} \overleftarrow{\partial}_{\xi^i} \overrightarrow{\partial}_{\xi^j} \right\}, \tag{A.1}$$

where $\omega_\xi^{ij}$ is the symplectic 2-form for the phase space coordinates $\xi$. One may express Eq. (A.1) as a gradient expansion with terms that are increasingly irrelevant for the low-energy effective field theory (EFT). To keep track of the order in the gradient expansion, we have introduced a dimensionless parameter $\hbar$ in Eq. (A.1), which we will set to one at the end to compare with the results in the main text, whereas the Planck constant and unit charge is set to one from the start. Ultimately, the gradient expansion is justified not because $\hbar$ is small, but because $B/k_F^2 \ll 1$. Under this convention, we have $B = \ell_B^{-2}$, where $\ell_B$ is the magnetic length.

Below, we consider $d = 2$, and define $\boldsymbol{\pi} = \boldsymbol{p} + \boldsymbol{A}$, $\boldsymbol{R} = \ell_B^2(\boldsymbol{p} - \boldsymbol{A}) \times \hat{z}$. For coordinates $\{x, y, p_x, p_y\}$, $\{x, y, \pi_x, \pi_y\}$ and $\{R_x, R_y, \pi_x, \pi_y\}$, $\omega$ reads, respectively,

$$\omega^{x,p} = \begin{pmatrix} 0 & 0 & 1 & 0 \\ 0 & 0 & 0 & 1 \\ -1 & 0 & 0 & 0 \\ 0 & -1 & 0 & 0 \end{pmatrix}, \qquad \omega^{x,\pi} = \begin{pmatrix} 0 & 0 & 1 & 0 \\ 0 & 0 & 0 & 1 \\ -1 & 0 & 0 & -B \\ 0 & -1 & B & 0 \end{pmatrix},$$

$$\omega^{R,\pi} = \begin{pmatrix} 0 & 1/B & 0 & 0 \\ -1/B & 0 & 0 & 0 \\ 0 & 0 & 0 & -B \\ 0 & 0 & B & 0 \end{pmatrix}. \tag{A.2}$$

Note that Eq. (A.1) is exact for $\{x, p\}$, $\{x, \pi\}$, and $\{R, \pi\}$ coordinates, but its validity to all orders in the Moyal product has not been studied well for generalized phase space coordinates projected to a single band, like Eq. (A.19) we define below.

The multi-orbital coadjoint-orbit action reads

$$S = \int dt \frac{d\xi}{(2\pi)^2} \mathrm{Tr}\left[ \widehat{f}_0(\xi) \star \widehat{U}^{-1}(\xi, t) \star i\partial_t \widehat{U}(\xi, t) - \widehat{f}(\xi, t) \star \widehat{H}(\xi) \right], \tag{A.3}$$

where we have used $\sqrt{\det \omega} = 1$ from Eq. (A.2). Here, we take Eq. (A.1) and (A.3) as the starting point, and derive the effective action projected onto a band with a Fermi surface for a generic single particle Hamiltonian $\widehat{H}(\xi)$. In particular, in the weak magnetic field limit $(Ba_0^2 \ll 1)$, we choose the gauge such that a multi-orbital Hamiltonian that couples to external electromagnetic fields can be written as $\widehat{H}(\xi) = \widehat{H}_0(\boldsymbol{p} + \boldsymbol{A}) + A_0(\boldsymbol{x}) = \widehat{H}_0(\boldsymbol{\pi}) + A_0(\boldsymbol{x})$, where $\boldsymbol{\nabla} \times \boldsymbol{A} = B\hat{z}, -\boldsymbol{\nabla} A_0 = \boldsymbol{E}$.

## A.1 Star diagonalization

As a notational convention, we use $\widehat{V}$ to denote a matrix $V$ in orbital basis. Under the Moyal star product, we define a unitary transformation $\widehat{V}(\xi) \star \widehat{V}(\xi)^\dagger = \widehat{V}(\xi)^\dagger \star \widehat{V}(\xi) = 1$ such that a Hermitian matrix $\widehat{H}(\xi)$ is *star-diagoanlized* [49, 68], defined as $\widehat{\mathcal{E}} = \widehat{V} \star \widehat{H} \star \widehat{V}^\dagger$. As the Moyal product is associative, the coadjoint orbit action Eq. (A.3) can be written as

$$S = \int dt \frac{d\xi}{(2\pi)^2} \operatorname{Tr}\left[\widehat{f}_{0,\nu}(\xi) \star \widehat{U}_\nu^{-1}(\xi, t) \star i\partial_t \widehat{U}_\nu(\xi, t) - \widehat{f}_\nu(\xi, t) \star \widehat{\mathcal{E}}(\xi)\right], \quad \text{(A.4)}$$

where $\widehat{f}_{0,\nu}(\xi) = \widehat{V} \star \widehat{f}_0(\xi) \star \widehat{V}^\dagger$, $\widehat{U}_\nu(\xi) = \widehat{V} \star \widehat{U}(\xi) \star \widehat{V}^\dagger$, and $\widehat{f}_\nu(\xi) = \widehat{U}_\nu(\xi) \star \widehat{f}_{0,\nu}(\xi) \star \widehat{U}_\nu^{-1}(\xi)$. Note that while $\widehat{f}_{0,\nu}$ and $\widehat{\mathcal{E}}$ are diagonal (defined as the band basis), $\widehat{U}_\nu$ and $\widehat{f}_\nu$ are not diagonal in general. However, the off-diagonal terms should be fast oscillating for band gap much greater than the EFT cutoff, and will be ignored hereafter. Projecting the action to the band (indexed by "$a$") at the chemical potential, the action reads

$$\begin{aligned} S_a = \int dt \frac{d\xi}{(2\pi)^2} \Big[ &f_{0,a}(\xi) \star U_{\nu,a}^{-1}(\xi, t) \star i\partial_t U_{\nu,a}(\xi, t) \\ &- U_{\nu,a}(\xi, t) \star f_{0,a}(\xi, t) \star U_{\nu,a}^{-1}(\xi, t) \star \mathcal{E}_a(\xi) \Big], \end{aligned} \quad \text{(A.5)}$$

where $f_{0,a}, U_{\nu,a}, f_{\nu,a}$ and $\mathcal{E}_a$ are the diagonal "$aa$" elements of the corresponding matrices in the band basis. In the $(\boldsymbol{R}, \boldsymbol{\pi})$ basis, and for Hamiltonian $\widehat{H}(\xi) = \widehat{H}(\boldsymbol{\pi})$, we can equate the expressions in the main text with expressions here by $f_0(\boldsymbol{\pi}) = f_{0,a}$, $U = U_{\nu,a}$, $\epsilon(\boldsymbol{\pi}) = \mathcal{E}_a$, and Eq. (A.5) can be equivalently expressed as Eq. (6) in the main text. At first sight, it is equivalent to the coadjoint orbit action for a single band Hamiltonian. However, note that $\mathcal{E}_a(\xi)$ is obtained from the star diagonalization of $\widehat{H}$, which is different from the "band dispersion" obtained via the usual matrix diagonalization. To relate the two, define a unitary transformation $\widehat{V}_0$ that diagonalize $\widehat{H}$, i.e. $\widehat{V}_0 \widehat{H} \widehat{V}_0^\dagger = \widehat{\mathcal{E}}_0$, such that

$$\widehat{V} = (1 + \widehat{V}_1 + \widehat{V}_2 + \ldots)\widehat{V}_0, \quad \text{where} \ \ \widehat{V}_n \sim \mathcal{O}(\hbar^n). \quad \text{(A.6)}$$

Similarly, $\widehat{\mathcal{E}} = \widehat{\mathcal{E}}_0 + \sum_{n=1}^\infty \widehat{\mathcal{E}}_n$ where $\widehat{\mathcal{E}}_n \sim \mathcal{O}(\hbar^n)$. As $\widehat{V}_0$ and $\widehat{\mathcal{E}}_0$ encodes all information in $\widehat{H}$, $\widehat{V}_n$ and $\widehat{\mathcal{E}}_n$ ($n \geq 1$) can be determined order by order in terms of $\widehat{V}_0$ and $\widehat{\mathcal{E}}_0$ given the symplectic 2-form $\omega^{ij}$.

This procedure has been introduced and applied in earlier studies [68] in terms of $\{\boldsymbol{x}, \boldsymbol{p}\}$ coordinates. Below, we apply the procedure for a set of generic phase space coordinates $\{\xi\}$ and obtain the leading order terms in the gradient expansion, i.e. $\widehat{V}_1$ and $\widehat{\mathcal{E}}_1$.

Using $\widehat{V} \star \widehat{V}^\dagger = \widehat{V}^\dagger \star \widehat{V} = 1$ and $\widehat{V}_0 \widehat{V}_0^\dagger = \widehat{V}_0 \widehat{V}_0^\dagger = 1$, we find

$$\widehat{V}_1 + \widehat{V}_1^\dagger + \frac{i\hbar}{2}\omega^{ij}\widehat{\mathcal{A}}_i\widehat{\mathcal{A}}_j = 0, \quad \text{(A.7)}$$

where

$$\widehat{\mathcal{A}}_i = i\widehat{V}_0 \partial_{\xi^i}\widehat{V}_0^\dagger = \widehat{\mathcal{A}}_i^\dagger, \quad \text{(A.8)}$$

is the *non-Abelian* phase-space Berry connection, satisfying $(\partial_i \widehat{\mathcal{A}}_j) - (\partial_j \widehat{\mathcal{A}}_i) = i[\widehat{\mathcal{A}}_i, \widehat{\mathcal{A}}_j]$. The last term in (A.7) is Hermitian, and thus $\widehat{V}_1$ can be expressed via Hermitian and anti-Hermitian parts as

$$\widehat{V}_1 = -\frac{i\hbar}{4}\omega^{ij}\widehat{\mathcal{A}}_i\widehat{\mathcal{A}}_j + \widehat{\mathcal{Y}}_1, \quad \text{where} \ \ \widehat{\mathcal{Y}}_1^\dagger = -\widehat{\mathcal{Y}}_1. \quad \text{(A.9)}$$

The correction to the energy eigenvalue, $\widehat{\mathcal{E}}_1$ can be obtained by expanding $\widehat{\mathcal{E}} = \widehat{V} \star \widehat{H} \star \widehat{V}^\dagger$ to the leading order in $\hbar$. We have

$$\widehat{\mathcal{E}} = (1 + \widehat{V}_1)\widehat{V}_0\left(1 + \frac{i\hbar}{2}\omega^{ij}\overleftarrow{\partial}_i\overrightarrow{\partial}_j\right)\left[\widehat{H}\left(1 + \frac{i\hbar}{2}\omega^{ij}\overleftarrow{\partial}_i\overrightarrow{\partial}_j\right)\widehat{V}_0^\dagger(1 + \widehat{V}_1^\dagger)\right] + \mathcal{O}(\hbar^2), \quad \text{(A.10)}$$

where we have made use of the associativity of the Moyal product. The left/right arrowed derivatives act on all terms to their left/right until stopped by an unpaired left/right square bracket. At order $\hbar$, we find

$$
\begin{aligned}
\widehat{\mathcal{E}}_1 &= \widehat{V}_1 \widehat{\mathcal{E}}_0 + \widehat{\mathcal{E}}_0 \widehat{V}_1^\dagger + \frac{i\hbar}{2} \omega^{ij} \left( i \widehat{\mathcal{A}}_i \widehat{V}_0 (\partial_j \widehat{H}) \widehat{V}_0^\dagger + \widehat{\mathcal{A}}_i \widehat{\mathcal{E}}_0 \widehat{\mathcal{A}}_j - i \widehat{V}_0 (\partial_i \widehat{H}) \widehat{V}_0^\dagger \widehat{\mathcal{A}}_j \right) \\
&= \widehat{V}_1 \widehat{\mathcal{E}}_0 + \widehat{\mathcal{E}}_0 \widehat{V}_1^\dagger - \frac{\hbar}{2} \omega^{ij} \{ \widehat{\mathcal{A}}_i, (\partial_j \widehat{\mathcal{E}}_0) \}_+ + \frac{i\hbar}{2} \omega^{ij} \{ \widehat{\mathcal{A}}_i, [\widehat{\mathcal{A}}_j, \widehat{\mathcal{E}}_0] \}_+ + \frac{i\hbar}{2} \omega^{ij} \widehat{\mathcal{A}}_i \widehat{\mathcal{E}}_0 \widehat{\mathcal{A}}_j \qquad \text{(A.11)} \\
&= \left[ \mathcal{Y}_1, \widehat{\mathcal{E}}_0 \right] - \frac{\hbar}{2} \omega^{ij} \{ \widehat{\mathcal{A}}_i, (\partial_j \widehat{\mathcal{E}}_0) \}_+ + \frac{i\hbar}{4} \omega^{ij} \left( \widehat{\mathcal{A}}_i \widehat{\mathcal{A}}_j \widehat{\mathcal{E}}_0 + \widehat{\mathcal{E}}_0 \widehat{\mathcal{A}}_i \widehat{\mathcal{A}}_j - 2 \widehat{\mathcal{A}}_i \widehat{\mathcal{E}}_0 \widehat{\mathcal{A}}_j \right).
\end{aligned}
$$

From the first to the second line, we have used $\widehat{V}_0 (\partial_i \widehat{H}) \widehat{V}_0^\dagger = \partial_i \widehat{\mathcal{E}}_0 - i [\widehat{\mathcal{A}}_i, \widehat{\mathcal{E}}_0]$. From the second to the third line, we have used Eq. (A.9). The dispersion projected to band "$a$" can be read off from the corresponding diagonal element. Noticing that $[\widehat{\mathcal{Y}}_1, \widehat{\mathcal{E}}_0]$ is an antisymmetric matrix, we have

$$
\begin{aligned}
\mathcal{E}_a &= \mathcal{E}_{0,a} - \hbar \omega^{ij} \widehat{\mathcal{A}}_i^{aa} (\partial_j \mathcal{E}_{0,a}) + \frac{i}{4} \hbar \omega^{ij} \left( \widehat{\mathcal{A}}_i \widehat{\mathcal{A}}_j \widehat{\mathcal{E}}_0 + \widehat{\mathcal{E}}_0 \widehat{\mathcal{A}}_i \widehat{\mathcal{A}}_j - 2 \widehat{\mathcal{A}}_i \widehat{\mathcal{E}}_0 \widehat{\mathcal{A}}_j \right)^{aa} + \mathcal{O}(\hbar^2) \\
&= \mathcal{E}_{0,a} - \hbar \omega^{ij} \widehat{\mathcal{A}}_i^{aa} (\partial_j \mathcal{E}_{0,a}) - \frac{i}{2} \hbar \omega^{ij} \left[ -\left( \widehat{\mathcal{A}}_i \widehat{\mathcal{E}}_0 \widehat{\mathcal{A}}_j \right)^{aa} + i \partial_i \widehat{\mathcal{A}}_j^{aa} \mathcal{E}_{0,a} \right] + \mathcal{O}(\hbar^2) \qquad \text{(A.12)} \\
&= \mathcal{E}_{0,a} (\xi^i + \hbar \omega^{ij} \mathcal{A}_j) - \frac{i}{2} \hbar \omega^{ij} \langle \partial_i v_a | (\widehat{H} - \mathcal{E}_{0,a}) | \partial_j v_a \rangle + \mathcal{O}(\hbar^2).
\end{aligned}
$$

To be concise, we have defined in the last line the intra-band (Abelian) phase-space Berry connection as $\mathcal{A}_i = \widehat{\mathcal{A}}_i^{aa}$. Eq. (A.12) is the main result of this subsection. It shows that there are two contributions to the dispersion at $\hbar$ order. The second term on the RHS of the first line shift the coordinate $\xi^i$, the third term modifies the spectrum. Below, we discuss two applications. *First*, we consider coupling the theory to an external perpendicular magnetic field. *Second*, we consider the effect of both magnetic and electric field.

## A.2 Coadjoint orbit EFT in an external magnetic field

In $\{x, \pi\}$ or $\{R, \pi\}$ coordinates, the Hamiltonian reads $\widehat{H}(\xi) = \widehat{H}(\pi = p + A)$. From Eq. (A.2), the modified dispersion up to order $\hbar$ reads

$$
\mathcal{E}_a(\pi) = \mathcal{E}_{0,a}(\pi_x - \hbar B \mathcal{A}_{\pi_y}, \pi_y + \hbar B \mathcal{A}_{\pi_x}) + \frac{i\hbar B}{2} \langle \partial_\pi v_a | (\widehat{H} - \mathcal{E}_{0,a}) \times | \partial_\pi v_a \rangle + \mathcal{O}(\hbar^2). \quad \text{(A.13)}
$$

Noting that the $\pi_{x,y}$ component of the phase-space Berry connection is just the momentum-space Berry connection in the main text, this is precisely Eq. (7) we derived in the main text (after setting the expansion parameter $\hbar = 1$). As was shown in the main text, the new coordinates in the first term lead to modified Jacobian and Moyal product, which is written in terms of the Berry curvature.

In the $\{x, p\}$ coordinates, we expect the same result. However, the calculation is more involved. The modified dispersion up to order $\hbar$ reads

$$
\begin{aligned}
\mathcal{E}_a(p + A) = &\; \mathcal{E}_{0,a}(p - \hbar \mathcal{A}_x + A(x + \hbar \mathcal{A}_p)) \\
&- \frac{i}{2} \hbar \left\{ \sum_{l=x,y} \left[ -\left( \widehat{\mathcal{A}}_{x_l} \widehat{\mathcal{E}}_0 \widehat{\mathcal{A}}_{p_l} \right)^{aa} + i \partial_{x_l} \mathcal{A}_{p_l} \mathcal{E}_{0,a} \right] - (x \leftrightarrow p) \right\} + \mathcal{O}(\hbar^2) \\
= &\; \mathcal{E}_{0,a}\left( p - \hbar \mathcal{A}_x - \frac{B}{2} (x + \hbar \mathcal{A}_p) \times \hat{z} \right) \qquad \text{(A.14)} \\
&- \frac{i}{2} \hbar \left\{ \sum_{l=x,y} \left[ -\left( \widehat{\mathcal{A}}_{x_l} \widehat{\mathcal{E}}_0 \widehat{\mathcal{A}}_{p_l} \right)^{aa} + i \partial_{x_l} \mathcal{A}_{p_l} \mathcal{E}_{0,a} \right] - (x \leftrightarrow p) \right\} + \mathcal{O}(\hbar^2) \\
= &\; \mathcal{E}_{0,a}(p + A - \hbar B \mathcal{A}_p \times \hat{z}) + \frac{i\hbar B}{2} \langle \partial_p v_a | (\widehat{H} - \mathcal{E}_{0,a}) \times | \partial_p v_a \rangle + \mathcal{O}(\hbar^2).
\end{aligned}
$$

By definition,

$$\widehat{\mathcal{A}}_{x_l} = i\widehat{V}_0(\boldsymbol{p}+\boldsymbol{A})\partial_{x_l}\widehat{V}_0^\dagger(\boldsymbol{p}+\boldsymbol{A}), \tag{A.15}$$

$$\widehat{\mathcal{A}}_{p_l} = i\widehat{V}_0(\boldsymbol{p}+\boldsymbol{A})\partial_{p_l}\widehat{V}_0^\dagger(\boldsymbol{p}+\boldsymbol{A}) = \widehat{\mathcal{A}}_{\pi_l}. \tag{A.16}$$

To obtain the last line in Eq. (A.14), we have used

$$\widehat{\mathcal{A}}_{x_j} = i\widehat{V}_0\partial_{x_j}\widehat{V}_0^\dagger = i\widehat{V}_0\partial_{p_i}\widehat{V}_0^\dagger\Big|_{\boldsymbol{p}+\boldsymbol{A}}\frac{\partial A_i}{\partial x_j} = -\frac{B}{2}\epsilon_{ij}\widehat{\mathcal{A}}_{p_i} = -\frac{B}{2}\epsilon_{ij}\widehat{\mathcal{A}}_{\pi_i},$$

and $\partial_{x_i}\widehat{\mathcal{A}}_{p_i} = \partial_{p_j}\widehat{\mathcal{A}}_{p_i}\frac{\partial A_j}{\partial x_i} = -\frac{B}{2}\epsilon_{ij}\partial_{p_i}\widehat{\mathcal{A}}_{p_j} = -\frac{B}{2}\epsilon_{ij}\partial_{\pi_i}\widehat{\mathcal{A}}_{\pi_j}$.

It is straightforward to check that Eq. (A.13) and (A.14) are equivalent up to $\mathcal{O}(\hbar)$. However, from this exercise, we see that in a homogeneous magnetic field, the calculation can be done more conveniently in the $\{\boldsymbol{x},\boldsymbol{\pi}\}$ or $\{\boldsymbol{R},\boldsymbol{\pi}\}$ phase space coordinates as long as the symplectic 2-form is properly chosen as in Eq. (A.2).

### A.3  Semiclassical Boltzmann equation in external electromagnetic fields

In this subsection, we demonstrate the band projection procedure in the presence of an additional static electric field, which is useful for, e.g., transport. In the presence of both electric and magnetic field, the Hamiltonian in Eq. (A.4) reads

$$\widehat{H}(\boldsymbol{\xi}) = \widehat{H}_0(\boldsymbol{\pi}) + A_0(\boldsymbol{x}), \quad \text{where } \boldsymbol{E} = -\nabla_{\boldsymbol{x}}A_0, B\hat{z} = \nabla_{\boldsymbol{x}}\times\boldsymbol{A}. \tag{A.17}$$

For simplicity, we use the $\{\boldsymbol{x},\boldsymbol{\pi}\}$ phase space coordinates. As the potential term $A_0(\boldsymbol{x})$ is diagonal in the band basis, the only nonzero phase space Berry connection is $\widehat{\mathcal{A}}_\pi = i\widehat{V}_0(\boldsymbol{\pi})\partial_\pi\widehat{V}_0^\dagger(\boldsymbol{\pi})$, where $\widehat{V}_0\widehat{H}_0(\boldsymbol{\pi})\widehat{V}_0^\dagger = \widehat{\mathcal{E}}_{\text{band},0}$ gives the spectrum in zero magnetic field. Following Eq. (A.12), the modified dispersion to band "$a$" reads

$$\begin{aligned}
\mathcal{E}_a &= \mathcal{E}_{0,a} - \hbar\omega^{ij}\mathcal{A}_i(\partial_j\mathcal{E}_{0,a}) - \frac{i}{2}\hbar\omega^{ij}\left(-(\widehat{\mathcal{A}}_i\widehat{\mathcal{E}}_0\widehat{\mathcal{A}}_j)^{aa} + i\partial_i\mathcal{A}_j\mathcal{E}_{0,a}\right) + \mathcal{O}(\hbar^2) \\
&= \widehat{\mathcal{E}}_{\text{band},0,a}(\boldsymbol{\pi} - \hbar B\mathcal{A}_\pi\times\hat{z}) + A_0(\boldsymbol{x} + \hbar\mathcal{A}_\pi) \\
&\quad + \frac{i\hbar B}{2}\langle\partial_\pi v_a|(\widehat{H}-\mathcal{E}_{0,a})\times|\partial_\pi v_a\rangle + \mathcal{O}(\hbar^2).
\end{aligned} \tag{A.18}$$

The first and second term in the last line shows that the phase space coordinate is shifted by the Berry connection, the third term is the energy shift from spontaneous orbital magnetization, same as Eq. (7) in the main text. Note that there is no energy shift due to coupling with electric-dipole for free fermion model when choosing the static electromagnetic $U(1)$ gauge, though interaction effects can introduce electric dipole terms [41,69].

To further simplify the expression, we can define the new phase space coordinate as

$$\boldsymbol{k} = \boldsymbol{\pi} - \hbar B\mathcal{A}_\pi\times\hat{z}, \qquad \boldsymbol{X} = \boldsymbol{x} + \hbar\mathcal{A}_\pi. \tag{A.19}$$

The modified dispersion in Eq. (A.18) can be expressed as

$$\mathcal{E}_a(\boldsymbol{X},\boldsymbol{k}) = \mathcal{E}_{\text{band},0,a}(\boldsymbol{k}) + A_0(\boldsymbol{X}) - \hbar B\mathcal{M}(\boldsymbol{k}) + \mathcal{O}(\hbar^2), \tag{A.20}$$

where $\mathcal{M}$ is the spontaneous orbital magnetic moment. The symplectic 2-form in the $\{\boldsymbol{X},\boldsymbol{k}\}$

basis to the leading order in $\hbar$ reads

$$
\omega^{X,k} = \begin{pmatrix} 0 & \hbar\Omega & 1-\hbar B\Omega & 0 \\ -\hbar\Omega & 0 & 0 & 1-\hbar B\Omega \\ -1+\hbar B\Omega & 0 & 0 & -B(1-\hbar B\Omega) \\ 0 & -1+\hbar B\Omega & B(1-\hbar B\Omega) & 0 \end{pmatrix} + \mathcal{O}(\hbar^2)
$$

$$
= \frac{1}{1+\hbar B\Omega} \begin{pmatrix} 0 & \hbar\Omega & 1 & 0 \\ -\hbar\Omega & 0 & 0 & 1 \\ -1 & 0 & 0 & -B \\ 0 & -1 & B & 0 \end{pmatrix} + \mathcal{O}(\hbar^2).
$$

(A.21)

We remind $\Omega(\boldsymbol{k}) = i\epsilon_{ij}\partial_{k_i}\mathcal{A}_{k_j} = i\langle\partial_{\boldsymbol{k}}v_a| \times |\partial_{\boldsymbol{k}}v_a\rangle$ is the single band Berry curvature. We see that indeed the Berry curvature is analogous to a $\boldsymbol{k}$-space magnetic field, although we caution that $\omega^{X,k}$ cannot define the Moyal product for $X, \boldsymbol{k}$, because careful treatments of higher order gradient expansions in $\hbar$ are needed. Nevertheless, $\omega^{X,k}$ defines the semiclassical Poisson bracket, by expanding Eq. (A.1) to order $\hbar$. Plugging it into the semiclassical Boltzmann equation $\partial_t f(X,\boldsymbol{k}) + \{f(X,\boldsymbol{k}), \mathcal{E}_a(X,\boldsymbol{k})\} = 0$, we reproduce the Berry curvature effects on semiclassical Boltzmann equation [70]

$$
\partial_t f + \frac{1}{1+\hbar B\Omega(\boldsymbol{k})} \left(\boldsymbol{v}(\boldsymbol{k})\cdot\partial_X f + \boldsymbol{E}\cdot\partial_k f + B\boldsymbol{v}(\boldsymbol{k})\times\partial_k f + \hbar\Omega\boldsymbol{E}\times\partial_X f\right) = 0,
\qquad \text{(A.22)}
$$

where $\boldsymbol{v}(\boldsymbol{k}) = \partial_{\boldsymbol{k}}(\mathcal{E}_{\text{band},0,a}(\boldsymbol{k}) - \hbar B\mathcal{M}(\boldsymbol{k}))$. Setting $\boldsymbol{E} = 0$, we obtain the semiclassical Boltzmann equation for cyclotron orbits, i.e. Eq. (8) in the main text.

## A.4 Gauge invariance

Note that the star-diagonalization and projection procedure should leave all physical observables invariant up to a "star" gauge transformation [38,68], which reads $\widehat{V} \to \widehat{V}' = \widehat{\Theta}_g \star \widehat{V}$, where $\widehat{\Theta}_g$ is a diagonal unitary matrix such that $\Theta_g \star \Theta_g^\dagger = \Theta_g\Theta_g^\dagger = \mathbb{1}$. Under this transformation, the star-diagonalization of $\widehat{H}$ becomes $\widehat{\mathcal{E}} = \widehat{V}\star\widehat{H}\star\widehat{V}^\dagger \to \widehat{\mathcal{E}}' = \widehat{V}'\star\widehat{H}\star\widehat{V}'^\dagger$, where $\widehat{\mathcal{E}}'$ remains diagonal but in general $\widehat{\mathcal{E}}' \neq \widehat{\mathcal{E}}$.

However, all physically observable effects are expressed in terms of the Berry curvature $\Omega$ and the orbital moment $\mathcal{M}$, which are indeed gauge invariant. To show it (at leading order in $\hbar$ in our procedure), we note that the gauge transformations of $\widehat{V}_n$ ($n = 0, 1, 2, \ldots$) can be obtained perturbatively using $\widehat{V}\star\widehat{V}^\dagger = 1$, and we find $\widehat{V}_0 \to \widehat{V}_0' = \Theta_g\widehat{V}_0$, $\widehat{V}_1 \to \widehat{V}_1' = \Theta_g\widehat{V}_1\Theta_g^\dagger - \frac{\hbar}{2}\omega^{ij}\partial_i\Theta_g\widehat{A}_j\Theta_g^\dagger$. The gauge field $\widehat{\mathcal{A}}$ transforms correspondingly as $\widehat{\mathcal{A}} \to \widehat{\mathcal{A}}' = i\widehat{\Theta}_g\partial_i\widehat{\Theta}_g^\dagger + \widehat{\Theta}_g\widehat{A}_i\widehat{\Theta}_g^\dagger$. The transformation for $\widehat{V}_0$ and $\widehat{\mathcal{A}}$ is reminiscent of the usual non-Abelian gauge transformation. It is then straightforward to show that both the Berry curvature for the projected band, written as $\Omega \propto \omega^{ij}\partial_i\mathcal{A}_j$, and the orbital moment of the projected band, written as $\mathcal{M} \propto -\frac{i}{2}\hbar\omega^{ij}\left[-\left(\widehat{\mathcal{A}}_i\widehat{\mathcal{E}}_0\widehat{\mathcal{A}}_j\right)^{aa} + i\partial_i\widehat{\mathcal{A}}_j^{aa}\mathcal{E}_{0,a}\right]$ are invariant under the "star" gauge transformation.

# B Anisotropic Fermi surfaces

In the main text, we focused on the simple case with an emergent isotropy near the FS. In this section we consider the general situation of anisotropic Fermi surfaces.

In this situation, polar coordinates for $\boldsymbol{k}$ space are no longer useful. Instead one needs to use a proper curvilinear coordinate system for each dispersion relation $\epsilon(\boldsymbol{k})$, which we sketch in Fig. 2. A natrual choice for the radial coordinate is proportional to $\epsilon(\boldsymbol{k})$ itself, which is a

constant on the FS. The second coordinate $\tilde{\theta} \in [-\pi, \pi\}$ acts as an angular variable on each contour. For simplicity of the Moyal bracket $\star = \exp\left[-iB\overleftarrow{\partial_{\boldsymbol{k}}} \times \overrightarrow{\partial_{\boldsymbol{k}}}/2(1+B\Omega)\right]$, we choose the second coordinate to be orthogonal to $\epsilon(\boldsymbol{k})$. The phase-space integration measure of $(\epsilon, \tilde{\theta})$ can be seen from the well-known relation

$$\int \frac{dk_x dk_y}{4\pi^2} \ldots = \int d\epsilon d\tilde{\theta} \, g(\epsilon, \tilde{\theta}) \ldots, \tag{B.1}$$

where

$$\int d\tilde{\theta} g(\epsilon, \tilde{\theta}) = g(\epsilon), \tag{B.2}$$

is the density of states determined from the dispersion $\epsilon(\boldsymbol{k})$ (not including corrections from Berry phases. [25])

One particular useful choice of $\tilde{\theta}$ is such that the distinct modes in the mode expansion

$$\phi_i(\tilde{\theta}) = q_i + p_i \tilde{\theta} + \sum_{n \neq 0} \frac{a_{i,n}}{\sqrt{|n|}} e^{in\tilde{\theta}}, \qquad \tilde{\theta} \in [-\pi, \pi), \tag{B.3}$$

are orthogonal, e.g.,

$$\int_{-\pi}^{\pi} d\tilde{\theta} g(\epsilon_F, \tilde{\theta}) e^{i(n-m)\tilde{\theta}} \sim \delta_{nm}. \tag{B.4}$$

Obviously, such a choice requires $g(\epsilon_F, \tilde{\theta})$ to be a constant, i.e., $g(\epsilon_F, \tilde{\theta}) = g(\epsilon_F)/2\pi$. This can be done as long as the FS is away from van Hove points. We denote the angular variable $\tilde{\theta}$ of this specific coordinate system $\theta$. We emphasize that this is not to be confused with the polar angle from $\Gamma$ point of the BZ.

In the $(\epsilon, \theta)$ coordinates, the phase-space integration measure and the Moyal star product is given by

$$\int \frac{dk_x dk_y}{2\pi B}(1+B\Omega) \ldots = \int \frac{d\epsilon d\theta}{B} g(\epsilon)(1+B\Omega) \ldots, \tag{B.5}$$

$$F \star G = F \exp\left[-\frac{iB}{4\pi} \frac{\overleftarrow{\partial_\epsilon} \times \overrightarrow{\partial_\theta}}{g(\epsilon)(1+B\Omega)}\right] G. \tag{B.6}$$

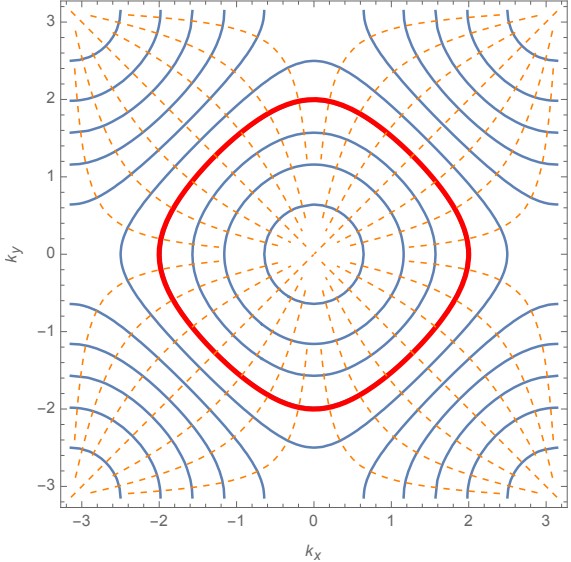

Figure 2: Orthogonal coordinates for a generic FS, denoted by the red thick line.

In this coordinate system, the effective action evaluates to $\sum_i^{N_\Phi} S_{\mathrm{w}}[\phi_i] + S_{\mathrm{cb}}[\phi_i]$, where

$$S_{\mathrm{cb}}[\phi] = \frac{1}{2} \int \frac{d\theta \, dt}{2\pi} \partial_\theta \phi \left[ \dot\phi - \frac{B}{2\pi g(\epsilon_F)(1 + B\Omega(\theta))} \partial_\theta \phi \right]. \tag{B.7}$$

Focusing on the $(q, p)$ modes of (B.3), integrating over $\theta$, and incorporating contributions from $S_{\mathrm{w}}$, we obtain

$$S_{\mathrm{zero}}[p, q] = \int dt \left[ \left( -\frac{\mathcal{A}_{\mathrm{FS}}}{2\pi B} - \frac{\gamma}{2\pi} \right) \dot q + p\dot q - \frac{\bar\omega_c}{2} p^2 \right], \tag{B.8}$$

where

$$\bar\omega_c = \frac{B}{2\pi g(\epsilon_F)} \left\langle \frac{1}{1 + B\Omega(\theta)} \right\rangle_{\mathrm{FS}}, \tag{B.9}$$

which is Eq. (13) in the main text. There $g(\epsilon_F)$ is denoted as $g(\epsilon_F, B)$ to emphasize its dependence on $B$ due to orbital magnetic moments.

## C Derivation of the modified Lifshitz-Kosevich formula

In this section, we detail the derivation of the modified Lifshitz-Kosevich formula from bosonization. We start with the part of the free energy from the zero modes

$$F_{\mathrm{zero}} = -N_\Phi T \log \left[ \sum_{p=-\infty}^{\infty} e^{-\beta \bar\omega_c (p + \Delta/2\pi)^2/2} \right], \tag{C.1}$$

where $\Delta = \mathcal{A}_{\mathrm{FS}}/B + \gamma$. Using the Poisson resummation formula, it can be rewritten as

$$\begin{aligned}
F_{\mathrm{zero}} &= -N_\Phi T \log \left[ \int dx \, e^{-\beta \bar\omega_c (x + \Delta/2\pi)^2/2} + 2 \sum_{n=1}^{\infty} \int dx \, e^{-\beta \bar\omega_c (x + \Delta/2\pi)^2/2} \cos(2\pi n x) \right] \\
&= -\frac{N_\Phi T}{2} \log \frac{2\pi T}{\bar\omega_c} - N_\Phi T \log \left[ 1 + 2 \sum_{n=1}^{\infty} \cos(n\Delta) e^{-2n^2 \pi^2 T/\bar\omega_c} \right].
\end{aligned} \tag{C.2}$$

Only the second term contains oscillation in $\Delta$, which is denoted as $F_{\mathrm{osc}}$ in the main text.

Note that from the definition of the Jacobi $\theta$-function [56]

$$\theta_3(z, q) = 1 + 2 \sum_{n=1} q^{n^2} \cos(2nz), \qquad |q| < 1, \tag{C.3}$$

we can rewrite $F_{\mathrm{osc}}$ as

$$F_{\mathrm{osc}} = -N_\Phi T \log \left[ \theta_3 \left( \frac{\Delta}{2}, q \right) \right], \tag{C.4}$$

where

$$\Delta = \frac{\mathcal{A}_{\mathrm{FS}}}{B} + \gamma, \qquad q = \exp\left( -\frac{2\pi^2 T}{\bar\omega_c} \right). \tag{C.5}$$

It is well-known that the Jacobi function $\theta_3$ has a product representation [56]:

$$\theta_3(z, q) = \prod_{n=1}^{\infty} (1 - q^{2n}) \prod_{n=1}^{\infty} \left[ 1 + 2q^{2n-1} \cos(2z) + q^{4n-2} \right]. \tag{C.6}$$

Using this, discarding nonoscillatory terms, we get

$$
\begin{aligned}
F_{\text{osc}} &= -N_\Phi T \sum_{n=1}^\infty \log\left[1 + 2q^{2n-1}\cos(\Delta) + q^{4n-2}\right] \\
&= -2N_\Phi T \operatorname{Re} \sum_{n=1}^\infty \log\left[1 + q^{2n-1}e^{i\Delta}\right].
\end{aligned}
\tag{C.7}
$$

Expanding the log and summing over $n$, we obtain

$$
\begin{aligned}
F_{\text{osc}} &= 2N_\Phi T \operatorname{Re} \sum_{n=1}^\infty \sum_{k=1}^\infty \frac{(-)^k}{k} q^{(2n-1)k} e^{ik\Delta} \\
&= 2N_\Phi T \sum_{k=1}^\infty \frac{(-)^k}{k} \frac{q^k}{1-q^{2k}} \cos(k\Delta).
\end{aligned}
\tag{C.8}
$$

Using (C.5), we get

$$
F_{\text{osc}} = \sum_{k=1}^\infty A_k \cos\left[k\left(\frac{\mathcal{A}_{\text{FS}}}{B} + \gamma\right)\right], \quad \text{where } A_k = \frac{(-)^k N_\Phi T}{k\sinh(2\pi^2 kT/\bar\omega_c)}.
\tag{C.9}
$$

This is the modified Lifshitz-Kosevich formula, Eq. (15) in the main text.

## C.1 Effects of $\phi^3$ terms in the action

In the main text, we expand the bosonic action up to quadratic order. For a parabolic dispersion, it was shown [30] that the action at $\phi^3$ order vanishes. However, in general for a lattice system such cubic terms do not vanish. In the main text, we argued that these terms are irrelevant perturbations, and hence only lead to higher-order corrections. In this section we demonstrate this by perturbatively evaluating the free energy with these terms. Without loss of generality, we assume the FS is isotropic.

At cubic order, the additional term in the action is [30]

$$
\begin{aligned}
S^{(3)}[\phi] &= \frac{1}{6} \sum_{K_i \in \text{mBZ}} \int \frac{dt\, d^2k}{2\pi B} \{\phi_i, f_0\}\{\phi_i, \{\phi_i, \epsilon(\boldsymbol{k})\}\} \\
&= \sum_{K_i \in \text{mBZ}} \int \frac{dt\, d\theta}{12\pi} \left[\frac{v(k)}{k}\right]'_{k=k_F} \frac{B^2}{k_F} (\partial_\theta \phi_i)^3.
\end{aligned}
\tag{C.10}
$$

For dHvA, we focus on the winding term $p\theta$ in the mode expansion (B.3). This leads to (neglecting Berry curvature terms, since we are looking for leading order corrections) the zero mode cubic action:

$$
S^{(3)}_{\text{zero}}[p,q] = \int dt \left[\frac{v(k)}{k}\right]'_{k=k_F} \frac{B^2}{6k_F} p^3 \equiv \frac{\lambda\bar\omega_c^2}{\epsilon_F} \int dt\, p^3,
\tag{C.11}
$$

where by dimensional analysis $\lambda$ is an $\mathcal{O}(1)$ number. For dispersion $\epsilon(\boldsymbol{k}) \propto k^\alpha$, $\lambda > 0$ when $\alpha > 2$, $\lambda < 0$ when $\alpha = 2$.

Together with the $S_{\text{zero}}[p,q]$ in Eq. (12) of the main text, we get after integrating out $q$:

$$
Z_{\text{zero}} = \sum_{p'\in\mathbb{Z}} \exp\left[-\frac{\beta\bar\omega_c}{2}\left(p' + \frac{\Delta}{2\pi}\right)^2 + \frac{\beta\lambda\bar\omega_c^2}{\epsilon_F}\left(p' + \frac{\Delta}{2\pi}\right)^3\right].
$$

From the fermionic perspective, for parabolic dispersion, the Landau levels are equally spaced, and $\bar{\omega}_c(p' + \Delta)^2/2$ is exactly the energy cost of adding $p'$ particles of momentum $K_i$ to the system [30]. With this insight, the physical origin of the $\omega_c^2(p'+\Delta)^3/\epsilon_F$ term is clear: it is due to the fact that the Landau levels are not equally spaced for a generic dispersion. The cubic term in the exponent is suppressed by $\omega_c/\epsilon_F \sim Bk_F^2 \ll 1$, and can be treated perturbatively.

Using Poisson resummation formula, the free energy is

$$
\begin{aligned}
F_{\text{zero}} =& -N_\Phi T \log \left\{ \int dx\, e^{-\beta\bar{\omega}_c(x+\Delta/2\pi)^2/2} \left[ 1 + \frac{\beta\lambda\bar{\omega}_c^2}{\epsilon_F}\left(x+\frac{\Delta}{2\pi}\right)^3 \right]\left[ 1 + 2\sum_{n=1}^\infty \cos(2\pi nx) \right] \right\} \\
=& -N_\Phi T \log \left\{ \int dy\, e^{-\beta\bar{\omega}_c y^2/2} \left( 1 + \frac{\beta\lambda\bar{\omega}_c^2}{\epsilon_F}y^3 \right)\left[ 1 + 2\sum_{n=1}^\infty \cos(2\pi ny - n\Delta) \right] \right\} \\
=& -\frac{N_\Phi T}{2} \log \frac{2\pi T}{\bar{\omega}_c} \\
& -N_\Phi T \log \left\{ 1 + 2\sum_{n=1}^\infty \left[ \cos(n\Delta) + \frac{\lambda\bar{\omega}_c}{\epsilon_F}\left( \frac{6n\pi}{\beta\bar{\omega}_c} - \frac{8n^3\pi^3}{\beta^2\bar{\omega}_c^2} \right)\sin(n\Delta) \right] e^{-2n^2\pi^2 T/\bar{\omega}_c} \right\}.
\end{aligned}
\tag{C.12}
$$

We see that the leading-order contribution from $\phi^3$ terms, which correspond to Landau level spacing variation, is temperature-induced phase shifts to all the $\cos(n\Delta)$ terms in the last line of Eq. (C.2). For $T \ll \bar{\omega}_c$, this correction is negligible. Indeed, when temperature is much smaller than the Landau level spacing at the Fermi level, dHvA is insensitive to the spacing between higher/lower Landau levels. For $\bar{\omega}_c \lesssim T$, a regime the oscillation amplitude is exponentially suppressed, this phase shift is of order $(\mathcal{A}_{\text{FS}}/B)(T/\epsilon_F)^2$, i.e. it leads to a temperature-dependent effective Fermi surface area $\mathcal{A}_{\text{FS}}(1 + \text{sgn}(\lambda)\mathcal{O}(T/\epsilon_F)^2)$. The same effect in 3d has been reported in systems with small $\epsilon_F$ [57]. This effect is distinct from the phase shift due to Berry phase effect, which is independent of $T$ and $B$. Furthermore, as long as $T \ll \sqrt{\epsilon_F\bar{\omega}_c}$, this phase shift is much smaller than that due to the Berry phase (as $\gamma = \mathcal{O}(B^0)$). Therefore they can be safely neglected for purposes of the current work.

## D  Relation between dHvA phase shift and anomalous Hall conductance

In this section we provide a more detailed derivation relating the dHvA phase shift to the anomalous Hall conductance $\sigma_{xy}^H$. As we discussed in the main text, the key ingredient of the proof is a UV/IR matching of the braiding algebra. We consider the "translation" operator (setting the lattice constant $a = 1$)

$$
\hat{\mathbb{T}}_{x,y} = \exp\left( i \sum_\pi \hat{N}_\pi \pi_{x,y} \right),
\tag{D.1}
$$

and evaluate the braiding algebra $\hat{\mathbb{T}}_x \hat{\mathbb{T}}_y \hat{\mathbb{T}}_x^{-1} \hat{\mathbb{T}}_y^{-1}$.

In the UV (lattice scale), we have

$$
\hat{\mathbb{T}}_{x,y} = \exp\left( i \sum_{i=1}^N \hat{\pi}_{i;x,y} \right),
\tag{D.2}
$$

where $N$ is the total particle number and $\hat{\pi}_i$ is the first-quantized kinetic momentum operator for the $i$-th particle. Using the fact that $[\hat{\pi}_x, \hat{\pi}_y] = -iB$, and the fact that $BL^2 = 2\pi$ (since $N_\Phi = 1$), we obtain

$$
\hat{\mathbb{T}}_x \hat{\mathbb{T}}_y \hat{\mathbb{T}}_x^{-1} \hat{\mathbb{T}}_y^{-1} = \prod_i^N \exp\left( \frac{2\pi i}{L^2} \right) = e^{2\pi i \nu},
\tag{D.3}
$$

where $v = N/L^2$.

In the IR (low energy limit), the translation operator is represented by operators that act near the FS. We have

$$\hat{\mathbb{T}}_{x,y} \sim \exp\left\{i \int_\theta \hat{n}(\theta)\left[k_{F;x,y}(\theta) \pm B\mathcal{A}_{y,x}(\theta)\right]\right\}, \tag{D.4}$$

where $k_{F;x,y}$ are the components of the Fermi momentum and we have used

$$k_{x,y} \equiv \pi_{x,y} \mp B\mathcal{A}_{y,x}. \tag{D.5}$$

Via the Baker-Campbell-Hausdorff formula $e^A e^B e^{-A} e^{-B} = e^{[A,B]}\cdots$, we then have

$$\hat{\mathbb{T}}_x \hat{\mathbb{T}}_y \hat{\mathbb{T}}_x^{-1} = \hat{\mathbb{T}}_y \exp\left\{-\int_{\theta,\theta'} [\hat{n}(\theta), \hat{n}(\theta')]\left(k_{F;x}(\theta) + B\mathcal{A}_y(\theta)\right)\left(k_{F;y}(\theta') - B\mathcal{A}_x(\theta')\right)\right\}. \tag{D.6}$$

Applying the Kac-Moody algebra

$$[\hat{n}(\theta), \hat{n}(\theta')] = \frac{-i}{2\pi}\delta'(\theta - \theta'), \tag{D.7}$$

and integrating by parts we get

$$\begin{aligned}
\hat{\mathbb{T}}_x \hat{\mathbb{T}}_y \hat{\mathbb{T}}_x^{-1} \hat{\mathbb{T}}_y^{-1} &= \exp\left[\frac{i}{2\pi}\oint \left(k_{F;x}(\theta) + B\mathcal{A}_y(\theta)\right) d\left(k_{F;y}(\theta) - B\mathcal{A}_x(\theta)\right)\right] \\
&= \exp\left[\frac{i}{2\pi}\oint k_{F;x}(\theta) dk_{F;y}(\theta)\right]\exp\left[\frac{iB}{2\pi}\oint \mathcal{A}(\theta)\cdot d\boldsymbol{k}_F(\theta)\right],
\end{aligned} \tag{D.8}$$

where we have kept only $\mathcal{O}(B)$ terms in the exponent, and integrated by parts again in the second factor of the last line. Preforming the integrals, we get

$$\hat{\mathbb{T}}_x \hat{\mathbb{T}}_y \hat{\mathbb{T}}_x^{-1} \hat{\mathbb{T}}_y^{-1} = \exp\left[i\frac{\mathcal{A}_{\mathrm{FS}}}{2\pi} + i\frac{B\gamma}{2\pi}\right], \tag{D.9}$$

which is Eq. (19) of the main text. Matching UV with IR, i.e., (D.3) with (D.9), using the Luttinger theorem and the Streda formula, we arrive at Eq. (20) of the main text:

$$\sigma^{\mathrm{H}}(q \to 0, \omega = 0) = \frac{\gamma}{4\pi^2} + \frac{\partial_B \mathcal{A}_{\mathrm{FS}}}{4\pi^2}. \tag{D.10}$$

## D.1 Validity in the presence of interaction effects

In the main text, our derivation for dHvA is explicitly performed for free fermions, and in the non-perturbative derivation above and in the main text, we have made use of Eq. (D.5), which in turn is obtained from free fermions. However, one can generalize both derivations to a Fermi liquid by incorporating Landau parameters and by interpreting $\epsilon(\boldsymbol{k})$ and $\gamma$ as dispersion and Berry phase of long-lived low-energy quasiparticles. Furthermore, the relation between the phase shift in dHvA and $\sigma^{\mathrm{H}}$ can also be applied to non-Fermi liquids. In fact, our argument can be made without referring to the Hamiltonian, interacting or free, at all!

To see this, we note that both the argument of the cosine terms in the Lifshitz-Kosevich formula (Eq. (15) of the main text) and the Kac-Moody algebra, which was used to compute $\sigma^{\mathrm{H}}$, come from the three $\sim \dot{\phi}$ terms in $S_{\mathrm{cb}} + S_{\mathrm{w}}$ (Eq. (10) of the main text.) These three terms are not independent from each other, but are derived from expanding the first term of the nonlinear action (Eq. (6) of the main text), which is a Wess-Zumino-Witten (WZW) term [30, 31]. Note that, being a first-order time-derivative, this term does not depend on the

Hamiltonian, and is thus robust against interaction effects. Without any information from the Hamiltonian, one can rewrite the WZW term in kinetic momentum $\pi$ coordinates as

$$S_{\text{WZW}} = \int_{t,\boldsymbol{R},\boldsymbol{\pi}} f_0(\boldsymbol{\pi}, B) U^{-1}(t, \boldsymbol{R}, \boldsymbol{\pi}) \star i \partial_t U(t, \boldsymbol{R}, \boldsymbol{\pi}), \tag{D.11}$$

where $f_0(\boldsymbol{\pi}, B)$ equals 0 or 1 on either side of the Fermi surface, specified by $\boldsymbol{\pi} = \boldsymbol{\pi}_F(B)$. Repeating our derivation in the main text, we get for the zero mode action (cf. Eq. (12) of the main text)

$$S_{\text{zero}}[p, q] = \int dt \left[ \left( -\frac{\mathcal{A}_{\text{FS}}^{\pi}(B)}{2\pi B} \right) \dot{q} + p\dot{q} - \cdots \right], \tag{D.12}$$

where $\mathcal{A}_{\text{FS}}^{\pi}(B)$ is the area of the FS in $(\pi_x, \pi_y)$ coordinates in a magnetic field, and for the braiding algebra (cf. Eq. (19) of the main text)

$$\hat{\mathbb{T}}_x \hat{\mathbb{T}}_y \hat{\mathbb{T}}_x^{-1} \hat{\mathbb{T}}_y^{-1} = \exp\left[ i \frac{\mathcal{A}_{\text{FS}}^{\pi}(B)}{2\pi} \right]. \tag{D.13}$$

Eq. (D.12) gives the phase shift in dHvA, and Eq. (D.13) gives the anomalous Hall conductance, and thus the two quantities are directly related.

Note that all of the above are done without explicitly referring to the Berry phase and the spontaneous magnetic moment; only the size of the FS in $(\pi_x, \pi_y)$ coordinates, which is well-defined even for non-Fermi liquids, matters. Of course, the amplitude $A_k$ of dHvA does depend on the Berry phase and the spontaneous moment separately, as we showed in the main text.

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
