# Peer review of "Berry Phase and Quantum Oscillation from Multi-orbital Coadjoint-orbit Bosonization"

_SciPost Physics, doi:SciPost Phys. 19, 078 (2025)_

## Round 1 · Referee Report · Anonymous (Referee 1) · 2025-6-25

Strengths

  1. The paper is concise and clear in its goal and exposition of the formalism it develops to describe an bosonized effective field theory for Fermi liquids in the presence of both a weak magnetic field and a Berry curvature.

  2. The modification of the Moyal product in the presence of both a Berry curvature and a weak magnetic field and the derivation of the effective field theory for free Fermi surfaces starting from the path integral via a gradient expansion of the Moyal product is a valuable addition to the literature.

Weaknesses

  1. The consequences of interactions is missing from the manuscript, which only deals with free Fermi surfaces, and this is explicitly noted by the authors in the paper.

  2. Relatedly, the bosonized effective field theory for free fermions without a magnetic field or Berry curvature is known to run into problems from UV/IR mixing upon quantizing the nonlinear terms in the gradient expansion. A discussion about whether these issues persist in this formalism, and whether they affect the nonperturbative results of the paper, would help tie any remaining loose ends.

Report

The paper is strong and easily meets the criteria for publication to SciPost.

Recommendation

Publish (easily meets expectations and criteria for this Journal; among top 50%)

  • validity: high
  • significance: good
  • originality: high
  • clarity: high
  • formatting: excellent
  • grammar: excellent

Author:  Yuxuan Wang  on 2025-07-16  [id 5646]

(in reply to Report 1 on 2025-06-25)

We thank the Referee for their positive evaluation of our work.

We thank the referee for their insightful question on possible UV/IR mixing issues in bosonized effective theory . The short answer is that these issues are resolved in the presence of a weak magnetic field. This point was addressed in a recent paper (Phys. Rev. B 112, 035113) by the two of us, and in the next version of the manuscript we will further clarify this issue.

The referee is correct that our calculation of the Lifshitz-Kosevich formula is based on a model for free fermions. However, we respectfully disagree with the Referee's comment that our paper does not address the consequences of interactions. In fact, in the abstract it is explicitly mentioned that "Beyond previously known results, we show that this phase shift holds even for interacting systems, in which the single-particle Berry phase is replaced by the static anomalous Hall conductance." This claim is backed up by Sec. 6 and Appendix D of the manuscript. To avoid future confusion, we plan to update the manuscript such that the applicability of our key results is presented more clearly.

---

## Round 1 · Referee Report · Anonymous (Referee 2) · 2025-7-31

Strengths

The paper clearly laid out a field theory framework to derive the de Haas-van Alphen effect in a Fermi liquid in the presence of a magnetic field and Berry curvature.

Weaknesses

The discussion on the effect of Berry curvature on the amplitude is not adequately addressed. As the authors point out, this effect could be compared with experimental results, yet such a comparison is missing in their work.

Report

The paper meets the standard for this journal, and I would recommend the publication if the requested changes are made appropriately.

Requested changes

In Eq. (16), the parameters $\lambda_1$ and $\lambda_2$ are not explicitly defined in the paper. Given the focus of this work, an explicit expression for these parameters—particularly their dependence on Berry curvature—would be helpful.

Recommendation

Publish (meets expectations and criteria for this Journal)

  • validity: high
  • significance: high
  • originality: high
  • clarity: high
  • formatting: perfect
  • grammar: perfect

Author:  Yuxuan Wang  on 2025-07-31  [id 5697]

(in reply to Report 2 on 2025-07-31)
Category:
remark

We are pleased with the positive evaluation of our work by the Referee. The requested changes, which will enable a direct comparison with experiments, will be implemented in the next version.

---

## Editorial Decision

published